# Mussel-Inspired Hydrogel Applied to Wound Healing: A Review and Future Prospects

**DOI:** 10.3390/biomimetics10040206

**Published:** 2025-03-26

**Authors:** Yanai Chen, Yijia Cao, Pengyu Cui, Shenzhou Lu

**Affiliations:** National Engineering Laboratory for Modern Silk, College of Textile and Clothing Engineering, Soochow University, Suzhou 215123, China; 20244015001@stu.suda.edu.cn (Y.C.); 20235215011@suda.edu.cn (Y.C.); 20234215009@suda.edu.cn (P.C.)

**Keywords:** mussel-inspired, hydrogel, wound healing

## Abstract

The application background of mussel-inspired materials is based on the unique underwater adhesive ability of marine mussels, which has inspired researchers to develop bionic materials with strong adhesion, self-healing ability, biocompatibility, and environmental friendliness. Specifically, 3, 4-dihydroxyphenylalanine (DOPA) in mussel byssus is able to form non-covalent forces on a variety of surfaces, which are critical for the mussel’s underwater adhesion and enable the mussel-inspired material to dissipate energy and repair itself under external forces. Mussel-inspired hydrogels are ideal medical adhesive materials due to their unique physical and chemical properties, such as excellent tissue adhesion, hemostasis and bacteriostasis, biosafety, and plasticity. This paper reviewed chitosan, cellulose, hyaluronic acid, gelatin, alginate, and other biomedical materials and discussed the advanced functions of mussel-inspired hydrogels as wound dressings, including antibacterial, anti-inflammatory, and antioxidant properties, adhesion and hemostasis, material transport, self-healing, stimulating response, and so on. At the same time, the technical challenges and limitations of the biomimetic mussel hydrogel in biomedical applications were further discussed, and its potential solutions and future research developments in the field of biomedicine were highlighted.

## 1. Introduction

With the survival of the fittest in nature, all kinds of species have experienced a long evolution, and their own structures can fully adapt to their respective environments. Among them, soft organisms such as marine mussels have evolved to firmly adhere to the surface of a variety of materials, such as rock and glass, in aquatic environments. Extensive studies have shown that this adhesion phenomenon is attributed to the catechol structure, 3,4-dihydroxyphenylalanine (DOPA), found in the adhesion-related proteins of mussel byssus [1,2,3]. Recently, a variety of new multifunctional materials have been developed by imitating the molecular structure of adhesion proteins and combining them with different materials [4]. These materials have become a prominent research focus in the field of biomimetic materials [5,6,7,8]. At present, mussel-inspired materials have made significant advancements in the fields of anti-pollution materials [9,10], flexible materials [11,12], hemostatic materials [13,14], adhesives [15,16], tissue repair materials [17,18,19], and other related fields [20,21,22].

As a promising soft material, hydrogels have been extensively investigated in biomedical research because of their unique three-dimensional network structure [23,24,25,26]. The adhesion ability of hydrogels is a significant challenge when they are used in the biomedical field [23,27,28]. In recent years, inspired by the chemistry of natural mussels, the emergence of mussel-inspired materials with excellent adhesion has inspired the development for the design of new self-adhesive hydrogels [29,30]. Especially in the field of wound repair, compared with traditional dressings, mussel-inspired hydrogels have superior wet adhesion properties, enabling firm adhesion to various substrates and biological tissue surfaces in a humid environment [25,31]. At the same time, these hydrogels have good biocompatibility, can provide a moist environment for wound healing, and reduce the risk of scar formation [24].

In this review, we aim to comprehensively explore the preparation of mussel-inspired hydrogels using various materials such as chitosan, cellulose, gelatin, hyaluronic acid, alginic acid, and their derivatives [32], as well as their advanced functionalities, including adhesion, antioxidant properties, and more (Figure 1). The key questions to be addressed include the following: (1) How can the composition and cross-linking strategies of these hydrogels be optimized to enhance their mechanical and functional properties? (2) What are the underlying mechanisms driving their superior adhesion and antioxidant capabilities? (3) How can their biocompatibility and biodegradability be further improved for specific clinical applications?

Catechol, a classical mussel-inspired structure, plays a pivotal role in these hydrogels. Its hydroxyl groups form stable hydrogen bonds with skin interfaces, significantly enhancing adhesion between the hydrogel and biological tissues. Additionally, the reductive nature of catechol hydroxyl groups enables them to scavenge oxygen free radicals through redox reactions, thereby mitigating oxidative damage. Natural polymer-based hydrogels also exhibit excellent biocompatibility, self-healing, and biodegradability, making them highly promising for biomedical applications, particularly in wound repair and tissue engineering. Despite these advantages, several challenges remain. For instance, the long-term stability and performance of these hydrogels under physiological conditions require further investigation. Moreover, the scalability of their production and their cost-effectiveness for clinical use require careful consideration. By addressing these questions and challenges, mussel-inspired hydrogels are expected to become indispensable materials in the medical field, paving the way for innovative therapeutic solutions.

## 2. Mussel-Inspired Hydrogel in Wound Repair

### 2.1. Chitosan Based Mussel-Inspired Hydrogel

Chitosan, as a natural polysaccharide containing both hydroxyl groups and amino groups [33], can form various derivatives with different structures and properties through acylation, carboxylation, hydroxylation, esterification, azide, salting, chelation, oxidation, grafting, and cross-linking [34]. It has a variety of excellent properties and wide application prospects. In the field of biology, it is a good choice for the skeleton of biomedical materials due to its biocompatibility and non-biotoxicity [35]. At the same time, it has the properties of acid resistance, being anti-ulcer, and promoting wound healing. The chitosan-based mussel-inspired hydrogel prepared by introducing a mussel-inspired structure usually has good adhesion, antibacterial properties, and cell proliferation.

Yang et al. [36] developed a photo-cross-linked, multifunctional, antibacterial, adhesive, antioxidant, and hemostatic hydrogel dressing based on polyethylene glycol monomethyl ether modified glycidyl methacrylate functionalized chitosan (CSG-PEG), methylacrylamide dopamine (DMA), and zinc ions. In mouse liver hemorrhage and tail amputation models, the CSG-PEG/DMA6/Zn hydrogel significantly reduced bleeding compared to the control group, demonstrating superior hemostatic efficacy (Figure 1A). The concentrations of DMA and zinc ions were varied to prepare the hydrogels, and they showed strong antibacterial activity against *S. aureus*, *E. coli*, and Methicillin-resistant Staphylococcus aureus (MRSA), with enhanced effects at higher zinc ion concentrations (Figure 1B). In a full-thickness skin defect model infected with MRSA, the CSG-PEG/DMA6/Zn hydrogel group achieved a significantly higher wound closure rate than the Tegaderm™ Film group, with nearly complete healing observed after 14 days (Figure 1C). These results highlight the hydrogel’s optimal wound healing performance.

The multifunctional CSG-PEG/DMA/Zn hydrogel integrates antibacterial, adhesive, antioxidant, and hemostatic properties, offering a promising solution for treating drug-resistant bacterial infections in wounds. Its antibiotic-independent antibacterial activity effectively inhibits a range of bacteria, including MRSA, with efficacy enhanced by higher zinc ion concentrations. The hydrogel also demonstrates superior hemostatic performance and accelerated wound healing in animal models, outperforming conventional dressings. However, challenges remain, such as optimizing the zinc ion concentration to balance antibacterial efficacy and potential toxicity. Further research is needed to assess long-term biocompatibility, mechanical stability, and scalability for clinical translation. Addressing these issues will be critical for advancing this innovative hydrogel toward practical applications.

Li et al. [37] designed a biomimetic hydrogel composed of caffeic acid-grafted chitosan (CHI-C), gallic acid-grafted chitosan (CSG), and oxidized microcrystalline cellulose (OMCC). The hydrogels exhibit blood-responsive gelation behavior with a synergistic combination of tissue adhesion, antibacterial properties, and tissue repair capabilities in a synergistic manner. The therapeutic efficacy of the CHI-C/CSG/OMCC ternary composite hydrogel was evaluated using endoscopy in a porcine model of acute gastrointestinal bleeding (Figure 2A,B). The results show that the hydrogel rapidly solidifies in the bleeding sites of the esophagus, stomach, and intestine, and effectively stopped bleeding without rebleeding. The effect of the CHI-C/CSG/OMCC hydrogel on full-thickness skin wound healing was evaluated by establishing a mouse skin defect model (Figure 2C,D). CHI-C/CSG and CHI-C/CSG/OMCC effectively gel in the wound area, adhere tightly to the skin, achieve hemostasis, and accelerate wound healing. The CHI-C/CSG/OMCC hydrogel, as a new biomimetic material synthesized by incorporating caffeic acid, gallic acid, and oxidized microcrystalline cellulose, demonstrates multifunctional properties, including rapid hemostasis, antibacterial activity, and the promotion of wound healing. In animal models, it exhibits significant hemostatic efficacy and excellent biocompatibility, highlighting its potential as a promising biomimetic material with significant potential for biomedical applications. However, the gelation rate may be influenced by factors such as bleeding volume and blood composition, necessitating further validation in diverse clinical scenarios to ensure consistency and reliability. Addressing these challenges will be crucial for translating this innovative hydrogel into practical clinical use.

Tian et al. [38] developed an adhesive, peptide-based, antibacterial, heat-sensitive hydroxybutyl chitosan (eLHBC) hydrogel inspired by mussels. HBC was modified with L-DOPA and ε-Poly-L-lysine (EPL) to obtain eLHBC hydrogels with tissue adhesion and antibacterial properties. eLHBC was prepared by covalently binding L-DOPA and EPL to HBC using EDC/NHS conjugation chemistry. The degree of substitution of eLHBC was controlled by adjusting the ratio of initial chemicals. eLHBC remained in solution at 4 °C and formed a hydrogel at 37 °C, serving as a 3D culture matrix for BMSCs (Figure 3A). The healing ability of BMSCs ⊂ eLHBC as a wound dressing was evaluated in a rat model. The BMSCs ⊂ eLHBC group had a wound closure rate of nearly 100% on day 15, while the other groups had lower closure rates (Figure 3B). The mussel-inspired eLHBC hydrogel exhibits enhanced wet adhesion and highly effective antibacterial activity as a wound dressing, preventing wound infection. The porous and interconnected structure of eLHBC provides an ideal 3D culture substrate for BMSCs. BMSCs encapsulated in eLHBC can secrete growth factors and promote the migration of fibroblasts. Animal experiments show that BMSCs ⊂ eLHBC reduces inflammation and significantly accelerates wound healing. Therefore, the eLHBC hydrogel is a promising candidate for promoting skin regeneration and treating skin wounds.

Although EPL (ε-polylysine) endows the hydrogel with antibacterial properties, its antimicrobial mechanism primarily relies on charge interactions, which may limit its efficacy against certain drug-resistant strains. Therefore, further optimization of antibacterial components or the incorporation of additional antimicrobial mechanisms could enhance the breadth of its antibacterial spectrum. Additionally, while the porous structure of the hydrogel facilitates cell growth and migration, the precise regulation of porosity and interconnectivity may require more refined design strategies to better meet the diverse requirements of tissue regeneration.

Hu et al. [39] developed a multifunctional, double cross-linked hydrogel through a Schiff base reaction between the amino groups (-NH_2_) in catechol–catechol adduction and chitosan quaternary ammonium salt (HTCC), and the aldehyde groups (-CHO) in oxidized dextran–dopamine (OD-DA). The dual cross-linking mechanism gives hydrogels excellent mechanical properties. Through the effective inclusion of silver nanoparticles (AgNPs) and the angiogenic drug deferoxamine (DFO), the hydrogel obtains antibacterial and angiogenic properties, respectively. The hydrogels, formed through catechol–catechol adduction and Schiff base reactions, exhibit self-healing properties. When cut in half and reconnected, the hydrogel pieces fully heal within 10 min of contact. The repaired hydrogel demonstrates sufficient mechanical strength to support its own weight without fracture, confirming its rapid self-healing ability (Figure 4A,B). In a rat wound model infected with *S. aureus*, the hydrogel@AgNPs&DFO group exhibited significantly faster wound closure compared to other groups, achieving a 79% closure rate within 7 days (Figure 4C). Double cross-linked hydrogels have rapid gelation and stable rheological properties. By introducing AgNPs and DFO, the hydrogels obtains significant antibacterial and pro-angiogenic properties. The hydrogel shows good antibacterial ability in vitro and in vivo, and has good cytocompatibility and blood compatibility. The hydrogel rapidly releases AgNPs to eliminate bacteria in the acidic microenvironment of the infected diabetic wound site, while releasing DFO to promote angiogenesis. The in vivo results demonstrated that the hydrogel@AgNPs&DFO group had faster wound healing, reduced inflammation, accelerated collagen deposition, and promoted angiogenesis. The new hydrogel is expected to be a promising wound dressing for the treatment of infected diabetic wounds and has great prospects for clinical translational application. However, despite its demonstrated efficacy and biocompatibility in experimental settings, the incorporation of silver nanoparticles (AgNPs), while significantly enhancing the antibacterial properties of the hydrogel, presents a critical challenge in ensuring their uniform dispersion within the hydrogel matrix. Inhomogeneous distribution may lead to suboptimal local antibacterial efficacy and even result in localized high concentrations, potentially causing cytotoxicity. Additionally, factors such as individual variability in complex clinical environments, cost-effectiveness, and potential long-term safety concerns must be carefully addressed. Therefore, while this research offers new hope for the treatment of diabetic wounds, it also faces significant challenges in being translated from the laboratory to clinical applications.

The preparation of the hydrogels is a complex and critical process that combines a variety of chemical modifications and cross-linking stages to endow the hydrogels with their unique properties. Chitosan and its derivatives are often modified with catecholic substances such as 3, 4-dihydroxyphenylacetic acid (DOPAC) [40,41]. Generally, DOPAC is dissolved first and reacted with EDC and NHS to activate carboxyl groups, then mixed with chitosan containing amino groups or its modified product, and reacts at a certain temperature to successfully introduce catechol groups [42]. The introduction of catechol groups can significantly enhance the adhesion and oxidation resistance of hydrogels. These groups can form hydrogen bonds and metal coordination interactions with various substrates, which is crucial for improving hydrogel performance. The cross-linked hydrogel network is usually constructed via a reaction with the cross-linking agent, forming the cross-linking based on the self-reaction of the polymer, or introducing metal ions for coordination [43,44]. In addition, usually such hydrogels will be combined with other components synergistically [45], and some studies have combined nanomaterials including nanoclays (such as Laponite) with chitosan hydrogels. The nanomaterials were uniformly dispersed in chitosan solution by ultrasonics or stirring and then modified and cross-linked. The addition of nanomaterials enhances the mechanical properties of hydrogels. Interactions between nanomaterials and polymer chains increase the rigidity and toughness of the network, potentially improving adhesion and antibacterial properties [46]. To enhance hydrogel functionality, bioactive ingredients such as polydopamine nanoparticles (PDA NPs) and antibacterial drugs are incorporated [47]. PDA NPs can be introduced into the hydrogel system through the Schiff base reaction of the quinone group on its surface with the amino group of chitosan or other polymers. These bioactive ingredients can give hydrogels antibacterial, antioxidant, or photothermal therapeutic functions, expanding the application potential of hydrogels in the biomedical field [48].

### 2.2. Gelatin Based Mussel-Inspired Hydrogel

Gelatin is a protein product obtained from the partial hydrolysis of collagen and has a complex molecular structure with a large number of hydroxyl, carboxyl, and amino groups in its molecules, which makes gelatin extremely hydrophilic and reactive [49,50]. Gelatin is widely used in the field of biomedicine because of its biocompatibility and biodegradability [51,52,53]. Specifically, it is modified with double bonds after methylallylation to make it have the characteristics of photo-cross-linking. Gelatin-based mussel-mimetic hydrogels have been widely used in the field of wound repair. They can be gradually degraded and absorbed in vivo without causing an immune reaction or inflammation.

Zhou et al. [54] have designed and prepared a gelatin-based adhesive (GMDA) hydrogel by mimicking the composition of mussel-adhesive proteins and the natural extracellular matrix (ECM). The hydrogel forms a double cross-linked network via horseradish peroxidase (HRP)/hydrogen peroxide (H_2_O_2_) and ultraviolet (UV) photo-cross-linking, giving the hydrogel adjustable mechanical properties and optimal elasticity (Figure 5A,B). In addition, these adhesive hydrogels can withstand a blood pressure of up to 250 mmHg (significantly higher than normal blood pressure) and showed superior in vivo hemostasis. Due to the residual H_2_O_2_ released after hydrogel formation, hydrogels exhibit remarkable antibacterial properties. More importantly, the GMDA hydrogel loaded with epidermal growth factor (EGF) demonstrated enhanced wound repair and promoted skin regeneration in a rat full-thickness skin defect model (Figure 5C). As a multifunctional biomaterial, the GMDA hydrogel developed in this study not only has excellent tissue adhesion and antimicrobial properties, but is also injectable and can adapt to defects of any geometric shape, providing a promising material for minimally invasive therapy. In addition, GMDA hydrogels have demonstrated effective hemostasis and tissue sealing properties in vivo, offering great potential for future use in a wide range of minimally invasive biomedical applications including hemostasis and wound infection healing. Although residual H_2_O_2_ imparts antibacterial properties to the hydrogel, excessive H_2_O_2_ may cause oxidative damage to surrounding healthy tissues.

Han et al. [55] prepared dopamine-modified gelatin (GelDA) and polydopamine-coated graphene oxide (pGO) and then designed GelDA/pGO hydrogels with adhesion, electrical conductivity, and antioxidant and antibacterial activity through oxidative coupling reactions between catechol groups. They also evaluated the ability of GelDA/pGO hydrogels to scavenge intracellular reactive oxygen species (ROS) through cell-based assays. In the experiment, L929 cells were stimulated to produce excess intracellular ROS using Rosup and then treated with the GelDA/pGO3 hydrogel (Figure 6A). The results show that the fluorescence intensity of intracellular ROS in the cells treated with the GelDA/pGO3 hydrogel was significantly reduced, and that the fluorescence intensity was similar to that in the untreated positive control group, indicating that the GelDA/pGO3 hydrogel had good intracellular ROS clearance ability. During burn wound healing, antioxidants facilitate tissue repair by neutralizing excess ROS and mitigating oxidative damage. The antioxidant properties of the GelDA/pGO hydrogels enable effective ROS removal, oxidative stress reduction, and the creation of a favorable microenvironment for wound healing (Figure 6B). This not only helps to reduce the inflammatory response, but also promotes cell proliferation and migration, speeding up wound healing. Specifically, in a rat burn model, the GelDA/pGO3 hydrogel significantly promoted wound healing, an effect that may be attributed to the hydrogel’s moisturizing and antioxidant properties. In particular, the GelDA/pGO3 + MP group exhibited the most pronounced wound healing effects, likely due to the sustained antibacterial activity of mupirocin. The GelDA/pGO hydrogel developed in this study as a multifunctional wound dressing not only has good tissue adhesion and antioxidant capacity, but also has electrical conductivity and photothermal antibacterial activity, providing a promising new approach for burn treatment. The versatility of these hydrogels allows them to effectively respond to the complex environment of burn wounds and promote wound healing. Although pGO provides the hydrogel with electrical conductivity and antibacterial activity, the biocompatibility of graphene oxide has remained a focal point of research. Further studies are needed to investigate the long-term cytotoxicity and tissue responses of pGO to ensure its safety in wound dressing applications.

Cheng et al. [56] used a method that mimics mussels’ strategy by chemically binding gelatin to dopamine groups to prepare hydrogel dressings with enhanced binding affinities. The dressing is loaded with cerium oxide nanoparticles (CeONs) and antimicrobial peptides (AMP) to achieve ROS clearance and antimicrobial properties. In in vivo experiments, the GelMA-DOPA-AMP-CeONs hydrogel not only accelerated wound healing (Figure 7A), but also promoted skin repair, reduced scar formation, and formed a microenvironment conducive to skin regeneration and remodeling under conditions of infection and inflammation through the antibacterial action of AMP and the ROS clearance action of CeONs (Figure 7B,C). This versatile hydrogel dressing has potential in wound management, especially in preventing infection, promoting rapid wound closure, and minimizing scar formation. By combining the antibacterial properties of AMP and the ROS clearance capabilities of CeONs, the dressing offers a promising solution for the treatment and management of chronic wounds. In addition, the ability to spray and adhesion of the dressing make it more convenient in clinical applications, reducing the frequency of dressing replacement and reducing the economic burden of treatment. Although cerium oxide nanoparticles (CeONs) exhibit excellent reactive oxygen species (ROS)-scavenging capabilities, their long-term stability within hydrogels requires further validation. Additionally, the activity of antimicrobial peptides (AMPs) may be compromised by factors such as the pH of the hydrogel, ionic strength, or interactions with other components.

Wang et al. [57] developed a mussel-inspired adhesive hydrogel wound dressing loaded with tetracycline hydrochloride (TH) to enhance skin regeneration. The addition of PDA significantly improved the adhesion of the hydrogel to a variety of substances including skin (Figure 8A), and compared with GE-PAM and PDA-GE-PAM, the TH-PDA-GE-PAM hydrogel had a larger antibacterial circle diameter on *E. coli* and *S. aureus*, demonstrating potent antibacterial activity (Figure 8B). In addition, in the rat model of the full-layer skin defect, the wound closure rate of the TH-PDA-GE-PAM hydrogel group was significantly higher than that of the blank group on day 14, showing the potential to promote complete skin regeneration (Figure 8C). This versatile hydrogel dressing has potential in wound management, especially in preventing infection, promoting rapid wound closure, and minimizing scar formation. By combining the adhesive properties of PDA with the antibacterial properties of TH, the dressing offers a promising solution for the treatment and management of chronic wounds. In addition, the ability to sprazy and adhesion of this dressing make it more convenient in clinical applications, reducing the frequency of dressing replacement and reducing the economic burden of treatment. Although the ability to spray the dressing improves its clinical convenience, further research is needed to ensure uniform coverage and adhesion strength after spraying, especially for wounds with complex shapes and sizes.

Gelatin-based mussel-inspired hydrogels exhibit unique advantages and significant potential in wound healing [58,59]. Their preparation methods are varied and fine, often using chemical modification and composite technology, such as grafting and cross-linking reactions. Their features include good biocompatibility, adjustable mechanical properties, excellent adhesion, and antibacterial properties. These hydrogels effectively enhance cell activity, control infection, accelerate hemostasis and wound healing, regulate inflammation, and promote tissue regeneration [41,60,61]. The innovative mussel-inspired design, multifunctional integration, and outstanding performance in various wound models offer new opportunities for wound treatment, positioning these hydrogels as promising materials for clinical wound management [62,63].

### 2.3. Hyaluronic Acid-Based Mussel-Inspired Hydrogel

Hyaluronic acid (HA) is an acidic mucopolysaccharide [64] that contains a large number of carboxyl groups and hydroxyl groups [65]. HA can promote cell proliferation and differentiation, remove oxygen free radicals, prevent and repair skin damage, and accelerate wound healing. Its unique physical and chemical properties give it significant advantages in moisturizing, lubricating, promoting wound healing, and preventing adhesions [66,67]. With the deepening of research, the combination of hyaluronic acid and mussel-inspired substances make it more widely used.

Li et al. [67] developed a hydrogel implantation system called BP-Ag@HA-DA-Plu, which incorporates a black phosphorus–silver nanocomplex (BP-Ag), dopamine-modified hyaluronic acid (HA-DA), and Pluronic^®^ F127 (Plu). Compared with the HA-DA-Plu hydrogel, the BP-Ag@HA-DA-Plu hydrogel shows remarkable photothermal effects under 808 nm NIR light irradiation and has increased photothermal effects when the temperature is increased by more than 20 °C (Figure 9A). After the tumor was established in the mouse model and surgically resected, different treatment methods were applied to different groups, including tumor resection only, tumor resection followed by the implantation of the BP@HA-DA-Plu hydrogel, tumor resection followed by the implantation of the BP-Ag@HA-DA-Plu hydrogel (Figure 9B), and the implantable BP-Ag@HA-DA-Plu hydrogel combined with near-infrared (NIR) light irradiation after tumor resection. The results show that tumor recurrence began on the 16th day after surgery in the group receiving resection and hydrogel implantation alone, and that the recurrent tumor tissue showed a rapid growth trend in the later period. The group receiving BP-Ag@HA-DA-Plu hydrogel implantation combined with multiple early photothermal interventions effectively inhibited tumor recurrence. In particular, the BP-Ag@HA-DA-Plu hydrogel combined with the NIR irradiation group showed no signs of recurrence on the 30th day after surgery, indicating that the hydrogel implantation system has great potential in preventing postoperative tumor recurrence. In addition, in the mouse wound infection model, the wound of the BP-Ag@HA-DA-Plu hydrogel combined with the NIR irradiation group was almost completely healed on the 14th day, while the wound area of the control group was still about 10% of the initial area (Figure 9C). In summary, the mussel-inspired “plug and play” hydrogel adhesive exhibits significant photothermal effects under 808 nm NIR irradiation, effectively killing residual tumor cells and bacteria. In mouse models, the hydrogel system significantly inhibited postoperative tumor recurrence and promoted wound healing, while showing good biocompatibility and safety, providing a new and efficient solution for clinical tumor resection. Although the ability to spray the dressing enhances its clinical convenience, further research is needed to ensure uniform coverage and adhesion strength after application, particularly for wounds with complex shapes and sizes.

An et al. [68] used chemical synthesis methods to introduce an aldehyde group into the HA backbone and then attach catechol to the HA through dual modes, including stable amide bonds and reactive secondary amine bonds. This design mimics the natural biochemical events of mussel foot proteins at different pH conditions, allowing the hydrogels to cross-link effectively under acidic, neutral, and alkaline conditions (Figure 10A). The untreated mice served as a control group by creating a 2 cm wide, full-layer incision in the skin of the mice and then sealing it with AH-CA hydrogel, HA-CA hydrogel, or fibrin-based adhesive. After 7 days, the wound areas were quantitatively measured using photographs. The AH-CA hydrogel group exhibited significantly smaller wound areas compared to other groups, demonstrating its potent wound-healing effects (Figure 10B). In the hemostasis experiment, the researchers used a mouse liver bleeding model to evaluate the in vivo hemostasis ability of the AH-CA hydrogel. Bleeding was induced by creating perforations in mouse livers, then the AH-CA hydrogel, HA-CA hydrogel, and fibrin-based adhesive were applied to the damaged site, respectively, and the weight of the filter paper that absorbed the blood was measured to calculate the amount of bleeding (Figure 10C,D). The results show that blood loss was significantly lower in the AH-CA hydrogel group than in the other groups, indicating its significant potential for emergency hemostasis, especially in clinical emergencies where rapid action and strong adhesion are required. In summary, the AH-CA hydrogels, inspired by mussel foot proteins, achieve effective cross-linking and adhesion across a wide pH range, offering significant potential for novel biomaterial development. The pH-universal catechol–amine chemical method provides a new strategy for the functionalization of HA hydrogels and is expected to be more widely used in the biomedical field. The cross-linking mechanism of AH-CA hydrogels involves multiple chemical bonds (e.g., amide and secondary amine bonds), which confer broad pH adaptability, but also increase the complexity of the synthesis process. For instance, precise control of the reaction conditions is required to ensure uniform cross-linking, potentially posing challenges for production efficiency and quality control.

Lv et al. [69] have developed a hydrogel composed of catechol-modified oxidized hyaluronic acid (OD), ε-poly-L-lysine (EPL), and Fe^3+^, prepared via a Schiff base reaction, metal chelation, cation–π, and electrostatic interactions, for the treatment of saltwater immersion wounds (Figure 11A). In terms of ROS removal, catechol groups in hydrogels are effective in reducing hydroxyl groups, which can effectively remove excess ROS in the wound microenvironment, reduce oxidative stress, and thus promote wound healing (Figure 11B). In terms of photothermal conversion, the hydrogel showed good photothermal response under 808 nm near-infrared light irradiation and was able to convert the absorbed light energy into heat energy, making the temperature rise rapidly (Figure 11C). The photothermal effect can not only be used for remote bacterial-killing therapy, but also speeds up microcirculatory blood flow, eliminates bacteria, reduces inflammation, and thus accelerates wound healing. In animal experiments, an infected wound model was created by seawater immersion. The OD/EPL@Fe hydrogel significantly promoted wound healing, reduced wound area, improved the healing rate, and achieved near-complete healing within 14 days (Figure 11D). On day 14, wounds in the hydrogel group exhibited a thicker epithelium and fewer inflammatory cells, indicating that the hydrogel accelerated the transition from the inflammatory phase to the repair phase. The OD/EPL@Fe hydrogel has potential in the treatment of seawater-soaked wounds, particularly in providing strong wet adhesion, antibacterial, antioxidant, anti-inflammatory, and pro-angiogenic activity. By mimicking the mussel’s adhesion mechanism and combining the dehydration effects with enhanced cohesion, the hydrogel achieves the effective treatment of seawater immersion wounds. In addition, the photothermal effects and antibacterial properties of hydrogels provide a new strategy for the treatment of seawater-soaked wounds. Fe^3^⁺ in this OD/EPL@Fe hydrogel may undergo redox reactions in physiological environments, affecting its photothermal performance and biocompatibility. It is necessary to investigate how to improve the stability of Fe^3^⁺ through chemical modifications or other methods.

Wang et al. [70] prepared a mussel-inspired dynamic cross-linked injectable hydrogel (DACS hydrogel) made from a mixture of dopamine-modified oxidized hyaluronic acid (DAHA) and carboxymethyl chitosan (CMCS) solutions. By adjusting the degree of substitution of DA and the concentration of raw material, the gelation time, rheology, and expansion characteristics of DACS hydrogels can be easily controlled. DACS hydrogels show good injectable and self-healing abilities to restore structure and function after damage and are suitable for use as tissue adhesives (Figure 12A). In addition, DACS hydrogels exhibit strong wet tissue adhesion to a variety of material surfaces, such as plastic, glass, and metal, approximately four times that of commercial fibrin adhesives (Figure 12B,C). The DACS hydrogel has potential as a multifunctional tissue-bonding material, especially in providing strong adhesion and self-healing and biodegradable properties. By simulating the adhesion mechanism of mussels and combining it with dynamic reversible cross-linking (Figure 12D), the hydrogel achieves the complete filling and rapid healing of irregular wounds. Although dynamic reversible cross-linking endows DACS hydrogels with excellent self-healing capabilities and adaptability, whether the dynamic cross-linking will gradually weaken in physiological environments during long-term use, thereby affecting the long-term performance of the hydrogels, requires further verification.

Mussel–hyaluronic acid hydrogels show significant advantages in the field of wound repair. The usual preparation methods include chemical modification cross-linking, such as the chemical modification of hyaluronic acid and its related polymers to introduce active groups, and then a cross-linking reaction to prepare the hydrogels. They also include physical composite assembly, which uses layer by layer self-assembly technology to combine multiple polyelectrolytes to form multilayer hydrogels [71,72,73,74]. A variety of functions such as good biocompatibility, tissue adhesion, antibacterial properties, self-healing, and adjustable mechanical properties are integrated into the hydrogel. These mussel-inspired hyaluronic acid hydrogels show great potential in the field of wound repair and provide new ideas and methods for the development of novel wound dressings, but further research and clinical trials are still needed to promote their clinical application [75].

### 2.4. Cellulose-Based Mussel-Inspired Hydrogel

Cellulose is the most abundant polysaccharide on earth [76] and one of the most abundant renewable resources in nature [77,78]. The sources of cellulose mainly include natural cellulose extracted from cotton, wood and hemp cellulose extracted from silk and wool, and bacterial cellulose extracted from microorganisms [79].

By mimicking the mussel’s adhesion mechanism, Xie et al. [80] constructed a natural hydrogel dressing consisting of catechol-modified carboxymethyl cellulose (CMC-DA) and tannic acid (TA) that has rapid shape adaptability, wet viscosity, and antioxidant capacity for irregular, deep, and frequently active diabetic wound repair. The CMC-DA/TA hydrogel demonstrated excellent antioxidant capacity in vitro. In cell models, L929 cells co-cultured with H_2_O_2_ showed high cell viability, indicating the hydrogel’s enhanced antioxidant activity (Figure 13A). As shown in Figure 13B, the hydrogel gained an excellent antibacterial ability against *E. coli*, *S. aureus*, and methicillin-resistant staphylococcus aureus (MRSA). In a rat model of liver injury, the CMC-DA/TA hydrogel showed better hemostasis than commercial gauze and gelatin sponges (Figure 13C). Most importantly, combined with the above advantages, the wound healing rate of the CMC-DA/TA hydrogel treatment was significantly faster than those of the gauze and 3M groups in the diabetic mouse model, and the results show that the inflammatory response was significantly reduced in the hydrogel group and the granulation tissue thickness and collagen deposition were significantly increased (Figure 13D,E). The bio-inspired CMC-DA/TA hydrogel has rapid shape adaptation, self-healing ability, strong wet adhesion, good biocompatibility, significant antioxidant capacity and antimicrobial properties, and the ability to promote diabetic wound healing in vivo. These properties position the CMC-DA/TA hydrogel as a promising candidate for managing chronic diabetic wounds. However, hydrogels still face challenges regarding preparation speed, uniform distribution, adhesion strength, adaptability, functional integration, long-term performance, and large-scale production.

Yang et al. [81] prepared a new type of hydrogel dressing by introducing a cationic polyelectrolyte brush into a polydopamine/polyacrylamide (PDA/PAM) hydrogel. The cationic polyelectrolyte brush consists of a polydiallyl dimethylammonium chloride (pDADMAC) brush grafted with bacterial cellulose (BC) nanofibers. This one-dimensional polymer brush has a rigid BC skeleton that enhances the mechanical properties of the hydrogel, while the positively charged quaternary ammonium groups provide long-lasting antimicrobial properties and promote the crawling and proliferation of negatively charged epidermal cells (Figure 14A). In addition, hydrogels are rich in catechol groups and can adhere to various surfaces to meet the needs of large movements in special parts. BCD/PDA/PAM hydrogels with different BCD ratios were co-cultured with mouse bone marrow-derived mesenchymal stem cells (BMSCs). The results show that BCD/PDA/PAM hydrogels not only had good cytocompatibility, but could also further promote cell proliferation (Figure 14B). The BCD/PDA/PAM hydrogel dressing exhibits high ductility, good adhesion, antibacterial properties, and biocompatibility, demonstrating its significant potential as a novel wound dressing. It is particularly suitable for dynamically active areas, semi-contaminated incisions, infected surgical wounds, and other wounds requiring frequent dressing changes. It is important to consider that cationic polyelectrolytes may undergo degradation or lose their activity in the physiological environment, affecting their antibacterial properties and cell proliferation promotion.

Mussel–cellulose hydrogels have great application potential in the field of wound healing, and their preparation methods include many innovative ones such as chemical modification and bioactive ingredient combination [82,83,84]. In terms of performance, they has strong adhesion to skin and tissue, can effectively inhibit a variety of bacteria to prevent infection, and have shown good hemostatic effects in animal models [81,85,86]. They have suitable mechanical properties and self-healing abilities to adapt to wound changes, as well as good biocompatibility to promote cell proliferation [87,88]. These characteristics mean that it is expected to become an efficient wound healing material.

### 2.5. Alginate-Based Mussel-Inspired Hydrogel

Alginate is a polyanionic polysaccharide extracted from brown algae [89,90]. Alginate itself is insoluble in water, but its sodium salts (such as sodium alginate) are soluble in water, non-toxic and non-stimulating to cells, and have good biological activity [91]. It has been widely used in the food industry, textiles, daily chemical industry, and biomedicine. Mussel-inspired hydrogels can be prepared in a variety of ways to improve their properties and applications [92,93].

Zhang et al. [93] prepared a novel nanocomposite hydrogel by oxidizing alginate, dopamine, and the antimicrobial peptide ε-polylysine (PL) to form a nanocomplex (ODP), which was subsequently cross-linked with acrylamide to produce the ODPA nanocomposite hydrogel. Three-dimensional culture techniques were used to evaluate the effects of ODPA hydrogels on the growth and survival of L929 fibroblasts. Laser scanning confocal microscopy revealed that after 1 and 5 days of culture, the cells not only proliferated on the surface of the hydrogel, but also penetrated deep into its interior. Additionally, the cell morphology transitioned from spindle-shaped to round or aggregated, demonstrating that the ODPA hydrogel exhibits excellent biocompatibility and supports cell attachment, proliferation, and migration (Figure 15A). It shows its potential application in tissue engineering and wound healing. The hemostasis of the ODPA hydrogel was evaluated using a rat hemorrhagic liver model, and the results show that the ODPA hydrogel completed hemostasis in approximately 30 s, and that the total blood loss was much lower than that of the PL-AM hydrogel and the blank group (Figure 15B). As a multi-functional wound dressing, the ODPA nanocomposite hydrogel inspired by mussels has multiple functions of adhesion, hemostasis, antibacterial activity, and promoting cell migration, which can cover the needs of the entire classical wound healing process. Therefore, ODPA hydrogels hold significant potential for practical applications in wound healing and tissue regeneration, offering a new strategy for the design of advanced multifunctional wound dressings in the future.

Chen et al. [94] prepared an ultra-tough self-healing hydrogel based on dopamine-grafted oxidized sodium alginate (OSA-DA) and polyacrylamide (PAM). The hydrogel is obtained through a combination of physical and chemical cross-linking, first combining the acrylamide (AM) monomer with the OSA-DA chain via a Schiff base reaction and then chemically polymerizing in the presence of an initiator (APS) to form the OSA-DA-PAM hydrogel. In the rat model, optical images of OSA-DA-PAM hydrogel-treated wounds on days 0, 5, and 15 showed that wounds healed significantly faster in the hydrogel-treated group than in the blank group. After 15 days, significant intercellular edema and excessive fibroblasts were observed in the blank group, while a large number of mature and compact collagen fibers were observed in the hydrogel-treated group (Figure 16A,B). This suggests that the healing process was faster in the hydrogel-treated group, especially when using hydrogels containing EGF. The new dual-network hydrogel based on OSA-DA and PAM has excellent self-healing capabilities and tissue adhesion, making it ideal for use in repairing wound tissue. In vivo rat experiments further confirmed that the OSA-DA-PAM hydrogel can promote tissue regeneration and accelerate the wound healing process, indicating that the hydrogel is a very suitable and promising biomaterial for wound dressing. Dopamine may undergo oxidation, affecting its adhesion properties and the overall stability of the hydrogel. Researchers need to study how to improve the stability of dopamine via chemical modification or other methods.

Deng et al. [95] prepared a novel antimicrobial hydrogel that uses aluminum ion (Al^3+^) and alginate–dopamine (Alg-DA) chains to be cross-linked with copolymer chains of acrylamide and acrylic acid (PAM) through triple dynamic non-covalent interactions including coordination, electrostatic interactions, and hydrogen bonding. Hydrogels are self-healing at room temperature without any additives (Figure 17A). Hydrogels exhibit remarkable antibacterial activity due to the synergistic effect of cationized, nanofibrillated cellulose (CATNFC), Alg-DA, and Al^3+^. Growth tests for Staphylococcus aureus (*S. aureus*) and Escherichia coli (*E. coli*) showed that the Alg-DA-CATNFC-PAM-Al^3+^ hydrogel was effective in inhibiting Gram-positive and Gram-negative bacteria (Figure 17B). The wound healing effect of hydrogels was evaluated using a rat skin incision model (Figure 17C). The results show that the wounds treated with the hydrogel did not become infected, while the control group developed bacterial infections the day after the injury. Hydrogel-treated wounds did not show tears or exposed tissue in the early healing stage, which is attributed to the good bioadhesion of the hydrogel. The new hydrogel has high adhesion, toughness, and cell affinity, and unlike previous antibacterial hydrogels, it shows excellent contact activity and antibacterial properties. The application of CATNFC not only endows hydrogels with broad-spectrum antibacterial activity, but also improves their mechanical properties. DN hydrogels exhibit ideal mechanical properties and self-healing abilities due to strong cross-linked networks and dynamic non-covalent bonds. In addition, the prepared hydrogels can be easily recycled and are pollution free. In vitro antimicrobial tests and cell affinity and wound-healing experiments consistently show that the hydrogel can be used as a comprehensive biomaterial to prevent bacterial infections and promote tissue regeneration. Given these outstanding properties, the hydrogel is considered to have high potential in skin wound repair. Al^3^⁺ may be toxic to cells, especially with prolonged exposure. Whether the dynamic non-covalent bonds will gradually weaken in the physiological environment may be a slight problem affecting the long-term performance of the hydrogel.

Through chemical modification (such as oxidation, grafting, etc.) and the incorporation of functional substances (such as dopamine, ZIF-8 nanoparticles, etc.), the hydrogel network is formed via the interactions of Schiff base bonds, hydrogen bonds, and ionic bonds, etc. Some hydrogel preparations also involve the polymerization induced by UV irradiation, with the properties of the hydrogel being regulated by adjusting the proportions of each component [96,97,98,99]. The prepared hydrogel has the advantages of good mechanical properties, strong adhesion, antibacterial, antioxidant, suitable swelling, and degradation properties, and having good biocompatibility [100,101]. It is safe in cell and blood experiments and can effectively accelerate wound healing, promote tissue regeneration, and reduce inflammation in animal experiments, providing a potential material and method for wound repair. It is expected to promote the development of clinical wound treatment [102].

### 2.6. Other Mussel-Inspired Hydrogels

In addition to chitosan, gelatin, hyaluronic acid, cellulose, and alginate, mussel-inspired hydrogels also utilize biological materials such as silk fibroin, agarose, pectin, collagen, or a variety of modified polymers to functionalize and prepare mussel-mimetic multifunctional hydrogels for wound repair [103].

Mussel-inspired hydrogels have demonstrated remarkable potential in the field of wound repair, with numerous studies conducted to explore their applications. Multiple studies have been carried out on them. In terms of preparation methods, they have their own characteristics. For example, a double-network structure (PEI−PAA/Alg) is constructed by combining polyethyleneimine-polyacrylic acid with sodium alginate through electrostatic interaction and coordination complexation, and dopamine is introduced to enhance the adhesion performance [104], or a temperature-sensitive poly(N-isopropylacrylamide) microgel (MR) is introduced into the poly(acrylic acid)-poly(acrylamide)-poly(dopamine) (PAAc−PAM−PDA) hydrogel system [105], etc., to achieve the optimization and regulation of hydrogel properties.

These hydrogels have significant advantages in wound repair. The PEI−PAA/Alg hydrogel exhibits good mechanical properties, universal adhesion, tissue adhesion, and biocompatibility, and can promote wound healing; the MR/PAAc−PAM−PDA hydrogel boasts rapid self-healing, high stretchability, and strong adhesion, and can adhere to a variety of material surfaces; the PSNC hydrogel based on polymerized sulfobetaine methacrylate (SBMA), N-(2-amino-2-oxyethyl)acrylamide (NAGA) [106], and 1-carboxy-N-methyl-N-di(2-methacryloyloxy-ethyl)methanaminium inner salt (CBMAX) has skin-like mechanical properties, long-lasting moisture retention, temperature tolerance, and on-demand adhesion, and can accelerate the healing of burn wounds without secondary damage [107]; the CoSt hydrogel made from collagen and starch mimics the biological adhesion mechanism and performs well in hemostasis and promoting wound healing; the agarose-polydopamine hydrogel (APG) improves its performance by introducing polydopamine and can accelerate the repair of skin defects; the hydrogel prepared from soy protein isolate (SPI) and polyphenols has good underwater adhesion, antibacterial properties, and biocompatibility [107]. These research findings provide strong theoretical and practical support for the clinical application of mussel-inspired hydrogels in wound repair, advancing the development of innovative wound repair materials [108,109,110].

## 3. Discussion

Mussel-inspired hydrogels, as an emerging class of biomaterials, have demonstrated significant application potential in the biomedical field, particularly in wound dressings, tissue engineering, and drug delivery [111,112,113,114,115], thanks to their unique properties such as adhesion, antibacterial activity, antioxidant capacity, and self-healing ability. These hydrogels mimic the natural adhesion mechanism of mussel foot proteins by leveraging the chemical properties of catechol-based compounds like dopamine, enabling them to maintain stability in various complex environments (e.g., wet, dynamic, or contaminated conditions) and form strong adhesion to tissue surfaces. This characteristic provides mussel-inspired hydrogels with distinct advantages in treating irregular wounds, deep wounds, and wounds in highly mobile areas, effectively reducing the risk of dressing displacement and offering a stable microenvironment for wound healing. By incorporating antimicrobial peptides, metal ions, or other antibacterial components, mussel-inspired hydrogels can effectively inhibit the growth of both Gram-positive and Gram-negative bacteria, even showing antibacterial effects against drug-resistant strains. This antibacterial capability not only helps to prevent wound infections, but also reduces the complications caused by infections, thereby accelerating the wound healing process. Additionally, the antioxidant components in the hydrogels can scavenge reactive oxygen species (ROS) in the wound microenvironment, reducing oxidative stress and further promoting tissue repair and regeneration.

Despite significant progress in laboratory research, mussel-inspired hydrogels still face several challenges. First, the stability of catechol-based compounds like dopamine in physiological environments is relatively poor, as they are prone to oxidation, which may affect the long-term performance and biocompatibility of the hydrogels. Second, the biodegradability of many hydrogels is insufficient, potentially leading to residual accumulation in the body, necessitating the further optimization of their degradation mechanisms. Furthermore, the limited antibacterial spectrum is another issue, as some hydrogels exhibit restricted inhibitory effects against specific drug-resistant strains or fungi, requiring the development of broader antibacterial strategies. Additionally, the synergistic interactions among the various functional components in multifunctional hydrogels are not yet fully understood, which may result in interference between certain functions.

Several companies have developed medical patches based on mussel-inspired adhesion technology for wound closure and repair. For example, Mussel Polymers has developed a mussel-inspired adhesive material that has entered clinical trials for surgical wound closure and chronic wound treatment. The production cost of mussel-inspired hydrogels remains high, primarily due to the complexity of their synthesis processes and the cost of raw materials (e.g., dopamine derivatives, chitosan). However, with the increasing adoption of biomimetic materials in the medical field, consumer acceptance of high-performance wound dressings is gradually rising. Future trends will focus on reducing production costs by improving synthesis processes and adopting more economical raw materials, thereby promoting their commercialization. Beyond wound repair, mussel-inspired hydrogels also hold broad application potential in tissue engineering, drug delivery, and medical device coatings, which will further expand their market applicability.

Research on mussel-inspired hydrogels will advance in the directions of multifunctional integration, smart responsiveness, biocompatibility optimization, and antibacterial mechanism innovation. Researchers will strive to develop hydrogels that integrate multiple functions, such as antibacterial, antioxidant, pro-angiogenic, anti-inflammatory, and tissue-repair properties, to meet the multifaceted requirements of complex wound healing. Simultaneously, by designing smart hydrogels that respond to environmental changes (e.g., pH, temperature, or enzyme activity), precise drug release and dynamic regulation can be achieved. Moreover, combining nanotechnology or novel antimicrobial peptides to develop hydrogels with a broader antibacterial spectrum and enhanced antibacterial efficacy will help address the growing issue of drug-resistant bacteria. To improve the biocompatibility and biodegradability of hydrogels, researchers will explore new chemical modification methods or incorporate natural biomaterials. The ultimate goal is to develop customized hydrogels tailored to individual patient needs, such as specialized dressings for diabetic ulcers or drug-resistant infections, thereby advancing their transition from the laboratory to clinical applications and providing more effective solutions for complex wound healing and tissue regeneration.

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
