# Peer review of "Mussel-Inspired Hydrogel Applied to Wound Healing: A Review and Future Prospects"

_biomimetics, 2025, doi:10.3390/biomimetics10040206_

Round 1
Reviewer 1 Report
Comments and Suggestions for Authors
The authors of the paper "Mussel-inspired hydrogel applied to wound healing: review and future prospects" have presented a decent review article. However, the most significant criticism is that a large portion of the article consists of overly detailed summaries of other studies without providing an analysis of these studies, expressing their expert opinion, or comparing the results of multiple works. As a result, I cannot recommend this paper for publication, as it does not meet the journal's requirements for the "Review" type of publication. Nevertheless, the authors have compiled a substantial amount of material, which needs to be systematized and supplemented with their own perspective on the functional properties of various mussel-inspired hydrogels. More detailed comments are provided below:
- I recommend expanding the abbreviations in the abstract.
- Line 53: Typo – "The" should be corrected.
- The text lacks references to studies. While these references are present in the figure captions, they must also be included in the main text. Additionally, it is necessary to clarify whether permission has been obtained to use images from other articles.
- The authors state: "In the review, the preparation of mussel-like hydrogel by various materials such as chitosan, cellulose, gelatin, hyaluronic acid, alginic acid and their derivatives [34], and explores its advanced functions such as adhesion effect, biocompatibility, antibacterial and antioxidant properties, and self-healing properties." However, the work lacks a systematic organization of information regarding these "advanced functions," such as adhesion, biocompatibility, antibacterial and antioxidant properties, and self-healing properties. Simply stating that a certain type of hydrogel has good adhesive properties (or others) does not constitute a review.
- Overall, a significant portion of the work does not qualify as a review article because the authors merely describe in detail the results of various research groups without providing sufficient analysis, their own opinion, or comparisons between the results of different studies. For example, lines 64–174 simply describe four articles. This is not a scientific review, as it lacks new information from the authors of the review article: their own opinion, analysis, or comparison of data. In contrast, lines 175–200 represent what should be included in a review article. This issue applies to the entire work. Approximately 70% of this review article consists of summaries of others' research without the authors' expert input.
- The authors have not included indents before new chapters and subchapters. It is necessary to increase the indent size to improve readability.
- The authors also need to systematize the results of their review. Specifically, they should provide a summary table comparing different types of hydrogels and their properties, enabling readers to understand the global differences in properties among various hydrogel types. This is, in fact, what the authors claimed they would do in their article.
- The "Discussion" section lacks any discussion of the review results presented by the authors. In this section, the authors describe potential applications, but these are presented in vague terms without specific details.
- Although the authors occasionally compare results or analyze studies in some paragraphs, they do not provide concrete conclusions on the topic. It is necessary to explicitly state the unique features of each type of hydrogel in terms of specific properties at the end of each subchapter. These conclusions should then be summarized in the "Discussion" section, accompanied by a summary table and an analysis of the underlying reasons for these results.
In my opinion, this article may be considered for publication by the journal only after it has been revised to meet the requirements for "Review"-type articles.
Reviewer 2 Report
Comments and Suggestions for Authors
I reviewed the manuscript titled 'Mussel-inspired hydrogel applied to wound healing: review and future prospects.' The idea of mussel-inspired hydrogels initially seemed interesting; however, the manuscript requires significant revision. Below are my notes:
The text lacks proper references. It is not sufficient to just mention "Yang et al.," "Li et al.," etc.; the full citation should be provided: [number]. I see the number in the figure description. But it must be marked also in the text.
Another important point is whether the authors obtained permission from the publishers to reproduce figures. The information about permission should be included.
Please check that all species names (e.g., S. aureus) are written in italics. The same about in vivo and in vitro.
Line 13. The sentence is not clear enough. Avoid using abbreviations in the abstract. Check the term bysin, did you mean byssus? Non-covalent forces – interactions seem more appropriate in this context.
Line 34. the catechol (also known as catechol)
Lines 85, 121 and further in the whole text. Decrypt abbreviations at their first mention.
Figure 1B. Since you are including this figure, you need to explain exactly what is depicted in it. It is unclear what PBS, 0, 3, 6, and 9 represent. What method did the authors use for this test? Are these bacteria that appear after 2 hours in hydrogels with zero zinc content and do not appear at zinc concentrations of 3, 6, and 9? The reader should not guess what this represents—you need to describe it clearly.
Line 108. It would be helpful to specify how the hydrogel was introduced into the internal organs of the mice. Was it during the surgery?
The figures should be inserted after they are mentioned in the text, not before.
Line 110. Consider the possibility of rephrasing: “the hydrogel rapidly formed gel”
Line 147. Neither white arrows nor red dots are impossible to determine in the figure. The resolution is too poor.
Check the superscripts in the formulae, e.g., line 151.
Fig. 4C and line 161. Hydrogel 1, hydrogel 2, etc, is not very informative. Clarify what 1-4 mean.
Line 365. NT is for Not Treated? Please, clarify in the text.
Line 460-462. The two sentences convey the same idea, leading to redundancy. It would be more efficient to combine or rephrase them to avoid repetition.
Line 462-463. Why is the first group considered natural while the others are not? Is bacterial cellulose not produced by nature? It would be clearer if 'substances' were replaced with a more specific term, such as 'fibers.' What is 'cotton wood'? Animal cellulose from silk and wool? Unfortunately, I don’t have access to [83], but if this was really published in Carbohydrate Polymers, I’d pay to see it…
For Figure 14A, additional clarification is needed regarding the method used. What type of Live/dead staining method was applied in the context of evaluating antibacterial activity?
Figure 14B requires a more detailed description. It is unclear what exactly should be observed in the image. The figure should specify which color is used to label live and dead cells, and any other relevant details such as the markers or fluorescence stains applied. As it stands, the figure is not informative enough to interpret the results accurately.
Line 513. Mussel-like?
Line 532. What is AM-hydrogel?
Line 581. PAM is for polyacrylamide (made only from acrylamide monomers), while you describe copolymer chains of acrylamide and acrylic acid. Check it please.
Line 585. Both strains were mentioned many times above.
Line 648. The statement "Biomimetic mussel hydrogels based on chitosan, gelatin, hyaluronic acid, cellulose, and alginate have made significant progress in the field of wound repair" suggests that the hydrogels themselves have achieved progress, which is not accurate, as progress in scientific fields is made by researchers. Additionally, the phrase "made significant progress" could be reconsidered, as these materials are still in the experimental stage and have not yet been applied in real clinical settings.
Line 648-655. 2 sentences are repeated twice.
The discussion is weak. There is no specificity, analysis, or comparison. If we simplify this discussion, it would boil down to saying that hydrogels made from chitosan, gelatin, hyaluronic acid, cellulose, and alginate are good, and in the future, these hydrogels will get even better. Examples of the future prospective are given but they lack a clear foundation.
The presented reference list shows a regional bias, which may create a false impression of research concentration in this field being exclusively in East Asia. This limits the perception of a broader international scientific picture. It is important to consider the works of researchers from other regions to form a more objective and comprehensive understanding of current trends and achievements in the field. Alternatively, it could be noted that the review focuses on research from a specific region.
Overall, the manuscript seems more like a summary of a number of the recent research papers on various wound-healing hydrogels. To become a scientific review, it should include the authors' insights, problems, and solutions. I haven't detected those. So, the value of this work lies in the authors gathering and grouping 5-8 examples for each material, adding that they are mussel-like hydrogels (I'm not sure the authors of the cited works were aware of this), and briefly and not always clearly describing the clinical effects of the hydrogels. It is not enough.
Reviewer 3 Report
Comments and Suggestions for Authors
This article provides a good review on Mussel-inspired hydrogel and their applications in wound dressings. The following improvements are required in the article:
Please highlight how the work advances or increments the field from the present state of knowledge and provide a clear justification for your work.
There are number of grammatical errors in the manuscript which need to be corrected.
The current production/consumption data on commercial Mussel-inspired hydrogel and their products should be included.
Different strategies to produce Mussel-inspired hydrogel are discussed in this article. A paragraph on the comparison of these strategies should be added.
Improve the literature survey and also add the important results in abstract section.
State the objective of the investigation along with clear research questions at the end of the introduction.
Provide suitable references for each figure.
The linking of different parts of the review should be improved.
The critical review or comments are missing the manuscript.
Important results should be summarized at the end in tabulated form.
Comments on the Quality of English Language
Can be improved
Round 2
Reviewer 1 Report
Comments and Suggestions for Authors
The work of the authors has undergone significant changes. The authors have made numerous revisions to the manuscript. However, several comments remain:
1. Line 125: An extra period is present.
2. A table or a concluding chapter, in which the authors conduct a comparative analysis of hydrogels applied to wound healing, is essential to significantly enhance the quality of the article. This is because there are specific parameters by which the authors evaluate the effectiveness of various hydrogels applied to wound healing. The inclusion of this information would allow readers to familiarize themselves with the key findings of a wide range of studies and efficiently select the most suitable material for their needs. If readers require further details, they can refer to the main body of the article. The absence of a conclusion or a summary table significantly diminishes the value of this paper.
Reviewer 2 Report
Comments and Suggestions for Authors
The authors improved the manuscript.
However, I categorically protest against the sentence that mentions "animal cellulose," "cotton wood and hemp" as the primary sources of "natural" cellulose, as well as "fungal cellulose" (lines 531–533). It is incorrect and misleading. As I wrote before, I do not have access to [83]; not because I can't find it on the internet, but because it is not open access. The part that is available for free does not contain this sentence. Whatever nonsense is written there — it is on the conscience of the editor and reviewers. I can assure you that no fungal cellulose, and certainly no animal cellulose, exists on this planet. Cellulose is a component of the plant cell wall. And yes, some bacteria can synthesize it too. Not animals, not fungi.
I am also capable of googling and have found this "cotton wood", which is a poplar. - But this is illogical. There obviously must be a comma between "cotton" and "wood" in your sentence — then it makes sense. Don't tell me that "cotton wood" is the main source of cellulose; it is ridiculous.
Please, find some other sources of information. There are millions of reviews on cellulose available for reading, other than [83]. Maybe even one not from East Asia will be found.
I mark it as major revision to be sure that this will be revised, this is the only claim.
Reviewer 3 Report
Comments and Suggestions for Authors
The authors have made the necessary changes suggested by the reviewers.
Author Response
In order to ensure the quality of the article, we have fully revised it. Thank you again for your review.